# Biochemical Pattern of Methylmalonyl-CoA Epimerase Deficiency Identified in Newborn Screening: A Case Report

**DOI:** 10.3390/ijns10030053

**Published:** 2024-07-18

**Authors:** Evelina Maines, Roberto Franceschi, Francesca Rivieri, Giovanni Piccoli, Björn Schulte, Jessica Hoffmann, Andrea Bordugo, Giulia Rodella, Francesca Teofoli, Monica Vincenzi, Massimo Soffiati, Marta Camilot

**Affiliations:** 1Division of Pediatrics, Santa Chiara General Hospital, APSS Trento, 38122 Trento, Italy; roberto.franceschi@apss.tn.it (R.F.); massimo.soffiati@apss.tn.it (M.S.); 2Genetic Unit, Laboratory of Clinical Pathology, Department of Laboratories, APSS Trento, 38122 Trento, Italy; francesca.rivieri@apss.tn.it; 3CIBIO—Department of Cellular, Computational and Integrative Biology, Università degli Studi di Trento, 38122 Trento, Italy; giovanni.piccoli@unitn.it; 4CeGaT GmbH Tuebingen, 72076 Tuebingen, Germany; bjoern.schulte@humangenetik-tuebingen.de (B.S.); jessica.hoffmann@humangenetik-tuebingen.de (J.H.); 5Inherited Metabolic Disease Unit, Pediatric Department, AOUI Verona, 37134 Verona, Italy; andrea.bordugo@asufc.sanita.fvg.it (A.B.); giulia.rodella@aovr.veneto.it (G.R.); 6Department of Mother and Child, The Regional Center for Neonatal Screening, Diagnosis and Treatment of Inherited Congenital Metabolic and Endocrinological Diseases, AOUI Verona, 37134 Verona, Italy; francesca.teofoli@aovr.veneto.it (F.T.); monica.vincenzi@aovr.veneto.it (M.V.); marta.camilot@aovr.veneto.it (M.C.)

**Keywords:** methylmalonyl CoA epimerase deficiency, newborn screening, case report

## Abstract

Methylmalonyl-CoA epimerase enzyme (MCEE) is responsible for catalyzing the isomeric conversion between D- and L-methylmalonyl-CoA, an intermediate along the conversion of propionyl-CoA to succinyl-CoA. A dedicated test for MCEE deficiency is not included in the newborn screening (NBS) panels but it can be incidentally identified when investigating methylmalonic acidemia and propionic acidemia. Here, we report for the first time the biochemical description of a case detected by NBS. The NBS results showed increased levels of propionylcarnitine (C3) and 2-methylcitric acid (MCA), while methylmalonic acid (MMA) and homocysteine (Hcy) were within the reference limits. Confirmatory analyses revealed altered levels of metabolites, including MCA and MMA, suggesting a block in the propionate degradation pathway. The analysis of methylmalonic pathway genes by next-generation sequencing (NGS) allowed the identification of the known homozygous nonsense variation c.139C>T (p.R47X) in exon 2 of the MCE gene. Conclusions: Elevated concentrations of C3 with a slight increase in MCA and normal MMA and Hcy during NBS should prompt the consideration of MCEE deficiency in differential diagnosis. Increased MMA levels may be negligible at NBS as they may reach relevant values beyond the first days of life and thus could be identified only in confirmatory analyses.

## 1. Introduction

The human methylmalonyl-CoA epimerase gene (MCE) (MIM# 251120) was characterized and localized on chromosome 2p13.3 in 2001 by Bobik and Rasche [1]. MCEE catalyzes the isomeric conversion between D- and L-methylmalonyl-CoA, a key intermediate along the conversion of propionyl-CoA to succinyl-CoA, a necessary component in the tricarboxylic acid (TCA) cycle (Figure 1).

The first confirmed cases of variants in the human MCE gene were described in 2006 [2,3]. In two patients, concomitant variations in the Sepiapterin reductase gene (SPR) sufficiently explained the clinical symptoms [3,4], while other cases have been described as asymptomatic [5]. Heuberger et al. [6] recently investigated a cohort of 150 individuals suffering from methylmalonic aciduria of unknown origin and identified pathogenic variants in MCE in 10 patients. In these patients, the clinical symptoms were variable but usually mild. Disease onset ranged from 1 month to 2.5 years of age; at least three patients presented symptoms following an intercurrent illness.

MCEE deficiency is not included in the newborn screening (NBS) panels, but it can be incidentally identified while investigating for methylmalonyl-CoA mutase deficiency (MUT) (MIM# 251000) and propionic acidemia (PA) (MIM# 606054).

Newborn screening for MUT and PA is technically feasible for measuring the levels of propionylcarnitine (C3). Given that abnormal C3 levels are not disease-specific, 2nd tier testing for the detection of total homocysteine (Hcy), methylmalonic acid, and 2-methylcitric acid (MCA) is required to exclude false positives [7]. No literature reports describe the NBS results for MCEE deficiency. Here, we describe the first case.

## 2. Materials and Methods

### 2.1. Biochemical Evaluation

A 3 mm punch of a blood spot sample was first processed using the NeoBase Non-derivatized MSMS kit (Perkin Elmer, Wallac Oy, Turku, Finland) and analyzed by multiple reaction monitoring (MRM) in a tandem mass spectrometer TQD detector (Waters, Milford, MA, USA) equipped with a positively charged ESI source to obtain the acylcarnitine and amino acid profiles.

Second-tier testing for elevated C3 levels was performed according to the literature [8]. Moreover, the urinary acid profile was obtained according to the literature [9]. The plasma acylcarnitine profile was obtained with an appropriately modified NeoBase Non-derivatized MSMS kit (Perkin Elmer, Wallac Oy). Briefly, 3.21 μL of plasma was used instead of blood spots; the solution with a labeled internal standard added to it was stored at −20 °C for 10 min for protein precipitation and centrifuged at 14,000 rpm for 10 min before injection. Urinary organic acids were extracted with ethyl acetate, derivatized with trimethylsilyl (TMS) derivative, and analyzed by capillary gas chromatography–mass spectrometry (GC-MS) analysis.

### 2.2. Genetic Analysis

Written informed consent was obtained for the collection and storing of the child and parents’ clinical data and the analysis of the blood samples. Genomic DNA was extracted from peripheral venous blood on EDTA. Sequencing libraries of the patient were prepared using the Twist enrichment workflow (Twist Bioscience, San Francisco, CA, USA) and a custom-designed enrichment probe set (CeGaT ExomeXtra 3.0). Paired-end sequencing was performed on a NovaSeq instrument (Illumina, San Diego, CA, USA). Trimmed raw reads were aligned to the human genome (hg19) with the Burrows–Wheeler Aligner. The average coverage on targets for exome analysis was 134×. Sequence variants were called (CeGaT stratacall), with a minimum variant allele frequency of 1.5%. The resulting variants were annotated with population frequencies from gnomAD (2.1/3.1) and an internal database (CeGaT), with functional predictions from dbNSFP, with publications from the Human Gene Mutation Database (HGMD) that were available at the time, and with transcript information from Ensembl, RefSeq, Gencode, and CCDS.

We sequenced a whole exome by NGS. The analysis was initially restricted to the following gene regions: PCCA, PCCB, and MCEE genes. For segregation analysis, only the relevant region of the MCEE gene (NM_032601.4) was amplified via the PCR and directly sequenced using flanking or internal primer pairs for both parents.

## 3. Results

Our female patient was born to non-known consanguineous parents after 40 weeks of gestation, an uneventful pregnancy, and normal vaginal delivery. The Apgar score was 10 in the 1st and 5th minutes of life. At birth, the child’s weight was 2935 g, her length was 47 cm, and her head size was 32.5 cm. She was fed breast milk.

At 48 h of life, a few drops of blood were obtained via heel prick to perform NBS. The DBS was sent to the Newborn Screening Laboratory of the University of Verona (Italy).

The patient was discharged after three days of life on breast milk.

The acylcarnitine profile and 2nd tier testing of the first DBS are reported in Table 1. The exam revealed elevated C3 (7.26 µmol/L; n.v. < 4.66) and MCA levels (3.4 µmol/L; n.v. < 1), with normal MMA levels (4 µmol/L; n.v. < 5).

The baby was admitted to our Pediatric Unit at 11 days of life for clinical evaluation and confirmatory analysis. The clinical examination was completely normal. No dysmorphic signs were present. Her blood gases, ammonia, lactate, and glucose levels were normal.

DBS, plasma, and urine were collected and analyzed. Elevated C3 (2.66 µmol/L; n.v. 0.21–2.5) and MCA (1.9 µmol/L; n.v. < 1) with normal MMA (2.3 µmol/L; n.v. < 5) were confirmed on DBS. The urinary organic acids profile showed mildly increased MMA, with 3-OH-isovaleric acid, and slightly altered levels of MCA and 3-hydroxypropionic acid. All results are reported in Table 2.

The slightly altered MMA levels and increased 3-OH-isovaleric acid levels in urine prompted us to analyze the PCCA, PCCB and MCE genes before the MUT gene.

No treatment was started.

Regular follow-up showed normal growth parameters, neurological patterns, and normal creatine kinase (CK), transaminases, blood gasses, ammonia, and lactate levels.

Regular weaning was performed.

She is now 12 months old. No acute metabolic decompensation has been observed. A normoproteic diet (protein 2 g/kg/day) has been suggested. Protein restriction has been proposed during intercurrent illness, although its efficacy is unknown in MCEE deficiency.

### 3.1. Genetic Analysis

Methylmalonic pathway gene analysis using NGS identified the homozygous nonsense variation c.139C>T, p.(R47X) in exon 2 of the MCE gene. Evidence in the literature suggests that the p.(R47X) truncating mutation results in an inactive MCEE [3].

### 3.2. The Pathway of Propionyl-CoA to Succinyl-CoA Metabolism

In humans, propionyl-CoA is the product of the catabolism of four amino acids (isoleucine, valine, methionine, threonine), odd-chain fatty acids, the side chain of cholesterol, and propionyl-CoA derivates generated by anaerobic bacterial fermentation. The mitochondrial enzyme propionyl-CoA carboxylase (PCC) catalyzes the conversion of propionyl-CoA into methylmalonyl-CoA, which enters the Krebs cycle via succinyl-CoA. Located at the center of this pathway, MCEE catalyzes the epimerization of D-methylmalonyl-CoA to form L-methylmalonyl-CoA, the substrate of MUT. Defects of PCC cause propionic acidemia (PA), while inborn errors of the MUT gene lead to methylmalonic acidemia (MMA). MMA is characterized by elevated methylmalonic aciduria (MMAuria). On the contrary, methylmalonic acid is not elevated in PA, allowing distinction between MMA and PA. Moreover, while methylcitrate (MCA) and 3-hydroxypropionic acid are usually detected in urine by GC/MS in both disorders, 3-OH-isovaleric acid, N-propionylglycine and N-tiglylglycine are detectable in PA only [7].

## 4. Discussion

We report the first NBS results of a case of MCEE deficiency characterized by a mild increase in C3 and MCA levels. The MMA level was within the reference range for DBS at birth, and afterward, only a mild urinary excretion of MMA was discovered.

The NBS results could suggest PA but the biochemical profile obtained in the confirmatory analysis, such as mildly elevated MMA levels in urine, was not supportive.

MMAuria is the most common finding described in patients with pathogenic variants in the MCE gene, and MCE deficiency has often been identified in cases of MMAuria [2]. Nevertheless, very few data are available on NBS results.

Lund et al. (2012) described three cases of MCEE deficiency, notwithstanding negative results at NBS [10]. One boy had an average level of C3 upon neonatal DBS sampling, but later metabolic investigations, prompted by the unexplained death of a younger brother, showed vitamin B12-non-responsive MMA. Two sisters, the elder presenting an overt metabolic decompensation at the age of two, both had negative results upon NBS. Instead, the metabolic profile indicated MUT, though the MMA in urine was only mildly elevated (<1000 μmol/mmol creatinine).

Gradinger et al. reported a case with persistent MMA found by NBS in a clinically normal patient without other biochemical details [5].

Beyond NBS, pathogenic variations in MCE have been identified in several cases of MMAuria [2,3,4,5,6,10,11,12,13], both persisting [3] and intermittent [12]. The majority of patients with MCEE deficiency carry the homozygous nonsense variation c.139C>T, p.(R47X) [2,3,4,5,11,12]. The missense changes p.Lys60Gln and p.Arg143Cys [5], and a splice-altering variation (c.379-644A>G) [11] have also been identified, but their functional and pathological relevance remains unclear.

Heuberger et al. [6] recently identified 10 patients carrying pathogenic variations in MCE. In nine patients, they identified the homozygous nonsense variation c.139C>T, p.(R47X). In one, they reported the novel missense change c.158T>G (p.Ile53Arg). Urinary MMA was present in all patients (range 143–594 mmol/mol creatinine) and elevated in 5/10 patients. Patients carrying the same homozygous nonsense variation c.139C>T, p.(R47X) documented in our case showed heterogeneous clinical findings from no symptoms to severe metabolic acidosis and hypoglycemia. Most patients presented with metabolic decompensation during intercurrent illness in the first years of life. Beyond elevated urinary MMA, in four cases, elevated C3, 3-hydroxypropionic acid and/or MCA in urine were documented during metabolic decompensation [6]. In our patient, we have no data on the biochemical findings during decompensation because no acute metabolic decompensation was observed.

Abily-Donval et al. [12] described a patient presenting the mild and intermittent urinary excretion of MMA; in this case, MMA excretion was not observed during illness. A similar outcome was reported in another case [13]. The authors underlined that the limited sensitivity of the current GC-MS analytical methods may explain the failure to identify urinary MMA. More sensitive methods for MMA analysis, such as stable isotope dilution, are suggested [12]. It is important to remark that methylmalonyl-CoA is an optically active molecule with two possible configurations. On the contrary, MMA does not have an asymmetric carbon atom. Therefore, the MMA that accumulates in epimerase deficiency is identical to that observed in MUT [3]. In addition to MMA, elevated levels of other metabolites suggesting a block in the propionate degradation pathway, such as MCA, C3, and 3-hydroxypropionate, have been documented in most patients [6] and in our case.

Unlike other defects of the propionate pathway, the MCEE defect does not respond to vitamin B12 supplementation. In fact, MCEE belongs to the glyoxalase gene family that consists of six structurally and functionally diverse enzymes with nearly identical metal-binding sites and a strong preference for divalent metal ions and not cobalamin [14].

Further studies are necessary to provide more information on the NBS results and the MMA and MCA excretion patterns in patients with MCEE deficiency.

## 5. Conclusions

The diagnosis of MCEE deficiency by NBS is challenging. Elevated concentrations of C3 with a slight increase in MCA and normal MMA require the consideration of MCEE deficiency in the differential diagnosis. The MMA levels may be negligible on NBS and could be detected only after the first days of life in the case of confirmatory analyses.

Our case contributes to the biochemical characterization of this rare disorder. Additional cases of MCEE deficiency are required to set the NBS values pinpointing this rare condition.

## Figures and Tables

**Figure 1 IJNS-10-00053-f001:**
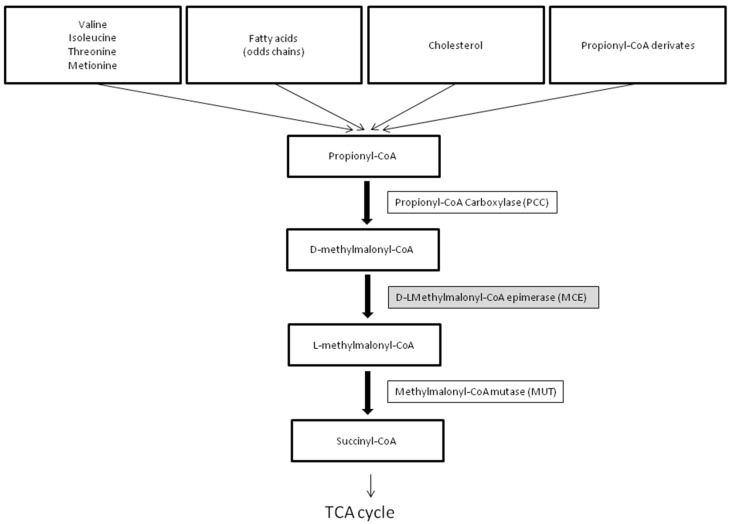
The pathway of propionyl-CoA to succinyl-CoA metabolism. See Section 3.2.

**Table 1 IJNS-10-00053-t001:** Analyses performed in DBS collected at 2 days of life.

	Primary Screening	Normal Range
C3	7.26	<4.66
C3/C2	0.31	<0.20
C16:1OH	0.11	<0.09
C0	17.9	6.29–47.5
	**2nd Tier Test**	**Normal Range**
MCA	3.4	<1
MMA	4	<5
Hcy	6.1	<11.9

All concentrations are expressed in µmol/L.

**Table 2 IJNS-10-00053-t002:** Diagnostic analyses performed at recall.

	DBS Results	Normal Range
C3	2.66	0.21–2.50
C2	5.09	3.31–30.12
C16:1OH	0.02	0.01–0.08
C0	21.12	7.26–54.05
MCA	1.9	<1
MMA	2.3	<5
Hcy	3	<11.9
	**Plasma Acylcarnitine Profile Results**	**Normal Range**
C3	4.63	0.17–2.82
C2	7.95	3.78–23.67
C16:1OH	0	0.00–0.02
C0	24.27	15.73–53.69
	**Urinary Organic Acids Profile Results**	**Normal Range**
MMA	58	<3
3-OH-isovaleric acid	117	<92
MCA, 3-hydroxypropionic acid, lactic acid	slightly altered	
N-propionylglycine N-tiglylglycine	Absent	Absent

The DBS and plasma acylcarnitines results are expressed in µmol/L. The urinary organic acid analysis is expressed in mmol/mol of creatinine.

## Data Availability

The raw data supporting the conclusions of this article will be made available by the authors on request.

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
