# Peer review of "Biochemical Pattern of Methylmalonyl-CoA Epimerase Deficiency Identified in Newborn Screening: A Case Report"

_2409-515X, 2024, doi:10.3390/ijns10030053_

Round 1
Reviewer 1 Report
Comments and Suggestions for Authors
Comments to Authors
This article an important case report of a case of MCEE deficiency detected in the neonatal period. I have some questions for the author regarding the homozygous nonsense mutation c.139C>T (p.R47X) identified in this case.
Although homozygous mutations of this variant have been reported previously in the literature, the elevation of 3-hydroxypropionic acid and other indices are low in the case you report. Therefore, you should include a discussion of your assessment of disease severity in relation to this case. There should also be a description of the management of the case, such as protein restriction. In this case, if more than one organic acid analysis was performed, please provide the data for the transitions.
Author Response
Reviewer 1#
This article an important case report of a case of MCEE deficiency detected in the neonatal period. I have some questions for the author regarding the homozygous nonsense mutation c.139C>T (p.R47X) identified in this case.
We are very thankful for the valuable comments of the reviewer. We greatly appreciate the opportunity to submit a revised version of our manuscript. The suggestions prompted us to improve our work.
Although homozygous mutations of this variant have been reported previously in the literature, the elevation of 3-hydroxypropionic acid and other indices are low in the case you report. Therefore, you should include a discussion of your assessment of disease severity in relation to this case.
Answer: thank you for your suggestion. We have added a statement on this point in the discussion (see page 6). Heuberger et al. (6) recently identified 10 patients carrying pathogenic variations in MCE. In nine patients, they identified the homozygous nonsense variation c.139C>T, p.(R47X), as reported in our case. Patients carrying the homozygous nonsense variation c.139C>T, p.(R47X) showed heterogeneous clinical findings from no symptoms to severe metabolic acidosis and hypoglycemia. Most patients presented with metabolic decompensation during intercurrent illness in the first years of life. Beyond elevated urinary MMA, in some cases elevated C3, 3-hydroxypropionic acid and MCA in urine were documented during metabolic decompensation (6).
Due to early diagnosis with NBS, as for other diseases which are on NBS panel, many patients are mildly affected or with no symptoms and remain so and is anyway very difficult to understand the degree of severity or a degree score for severity. The assessments in our patient were all made in a complete asymptomatic state and never during decompensation which up to now have never occurred. So the clinical severity score is a an ongoing research in which long term follow up will give us very important information not only for our patient but even for other patients and families.
There should also be a description of the management of the case, such as protein restriction.
Answer: thank you for your suggestion. We have added a statement on this point in the case description (see page 4). Normoproteic diet (protein 2 g/Kg/day) has been suggested. Protein restriction could be a therapeutic option during intercurrent illness, although its efficacy is unknown in MCEE deficiency.
Your question underscores the difficult task we have as clinicians when we have to decide which treatment we should use (aggressive protein restriction or free normal diet). Future data and new cases will hopefully improve our practice.
In this case, if more than one organic acid analysis was performed, please provide the data for the transitions.
Answer: A single urine collection was performed, because the meaning of the urinary organic acid analysis is mainly diagnostic. However, we plan to collect a new urine sample for organic acid analysis during intercurrent illness.
Reviewer 2 Report
Comments and Suggestions for Authors
This is a case report on a methylmalonyl-CoA epimerase (MCEE) deficiency that was detected through newborn screening (NBS). As the disease can be missed by the present tandem mass spectrometry-based NBS test using C3 as a primary marker, this report will be a caution for readers to bear it in mind as a candidate for differential diagnosis of C3-positive cases. The followings are my points to require the authors to revise.
Introduction/Discussion
It is difficult to make out clinical pictures of MCEE deficiency. It is advisable to add a table that lists clinical symptoms, biochemical data including, if any, NBS results of MCEE deficiency patients described in previous reports.
Methods for genetic analysis
I wonder why the analysis was restricted to PCCA, PCCB and MCEE. As potential causes for C3-positive in NBS and elevation of MMA in urine, genes for methylmalonyl-CoA mutase and proteins in the pathway of cobalamine metabolism should be analyzed.
Results
The information included in Table 1 and 2 must be presented in the main text, and appropriate titles must be added to these tables.
Author Response
Reviewer 2#
This is a case report on a methylmalonyl-CoA epimerase (MCEE) deficiency that was detected through newborn screening (NBS). As the disease can be missed by the present tandem mass spectrometry-based NBS test using C3 as a primary marker, this report will be a caution for readers to bear it in mind as a candidate for differential diagnosis of C3-positive cases. The followings are my points to require the authors to revise.
We are very thankful for the valuable comments of the reviewer. We greatly appreciate the opportunity to submit a revised version of our manuscript. The suggestions prompted us to improve our work.
Introduction/Discussion
It is difficult to make out clinical pictures of MCEE deficiency. It is advisable to add a table that lists clinical symptoms, biochemical data including, if any, NBS results of MCEE deficiency patients described in previous reports.
Answer: thank you for your suggestion. We added a statement in the discussion on heterogeneous clinical and biochemical findings of MCEE deficiency (see page 6).
Methods for genetic analysis
I wonder why the analysis was restricted to PCCA, PCCB and MCEE. As potential causes for C3-positive in NBS and elevation of MMA in urine, genes for methylmalonyl-CoA mutase and proteins in the pathway of cobalamine metabolism should be analyzed.
Answer: The initially ordered analysis included only the genes PCCA and PCCB. Then, we expanded the analysis to MCEE since a differential diagnostic screening approach should include variant in this gene.
Results
The information included in Table 1 and 2 must be presented in the main text, and appropriate titles must be added to these tables.
Answer: thank you for your suggestion. We added title tables. Data of table 1 and 2 are now presented in the text.
Reviewer 3 Report
Comments and Suggestions for Authors
This paper describes the NBS diagnosis of a child with MCEE deficiency. It documents the values of C3, MCA and MMA in the first child ever diagnosed via NBS
Material and methods are relatively clearly described, though it would be relevant to spell out what is meant by first-tier and second-tier analyses: the first-tier analyses are not described and concerning the second-tier it seemingly includes urinary and plasma analyses, which belongs to follow-up analyses. It becomes clearer in the tables, but should be described better in methods.
When for sequencing it is said that “Analysis was restricted …” – what does this imply? A virtual panel on a NGS platform? Or?
In results section: Apgar scores is normally written scores/minutes, but here is quoted e.g. “10/10 at 1st minute”, which I think would normally be written 10/1?
Variant should be written c.139C>T, p.(R47X).
Discussion is adequate and clear. I think it should be discussed whether MCEE deficiency should be on the NBS panel – does it fulfill WJ-criteria?
All in all, I think the case add important knowledge about the NBS diagnosis of MCEE deficiency.
Comments on the Quality of English LanguageAdequate - only few errors concerning use of definite article
Author Response
Reviewer 3#
This paper describes the NBS diagnosis of a child with MCEE deficiency. It documents the values of C3, MCA and MMA in the first child ever diagnosed via NBS
We are very thankful for the valuable comments of the reviewer. We greatly appreciate the opportunity to submit a revised version of our manuscript. The suggestions prompted us to improve our work.
Material and methods are relatively clearly described, though it would be relevant to spell out what is meant by first-tier and second-tier analyses: the first-tier analyses are not described and concerning the second-tier it seemingly includes urinary and plasma analyses, which belongs to follow-up analyses. It becomes clearer in the tables, but should be described better in methods.
Answer: thank you for your suggestion. We have added a statement on the meaning of 2nd tier testing in the introduction (see page 2). Moreover, we better clarified the results in the tables and the methods.
When for sequencing it is said that “Analysis was restricted …” – what does this imply? A virtual panel on a NGS platform? Or?
Answer: Yes, we sequenced a whole exome by NGS and virtually restricted the analysis to relevant genes. We better clarified this point in the methods.
In results section: Apgar scores is normally written scores/minutes, but here is quoted e.g. “10/10 at 1st minute”, which I think would normally be written 10/1?
Answer: we apologize for the mistake. The sentence has been rewritten.
Variant should be written c.139C>T, p.(R47X).
Answer: we apologize for the mistake. The variant has been rewritten.
Discussion is adequate and clear. I think it should be discussed whether MCEE deficiency should be on the NBS panel – does it fulfill WJ-criteria?
All in all, I think the case add important knowledge about the NBS diagnosis of MCEE deficiency.
Answer: thank you for your positive comment. A dedicated test for MCEE deficiency is not included in the newborn screening (NBS) panels but it can be incidentally identified when investigating methylmalonic acidemia and propionic acidemia. Our report will be a caution for readers to bear it in mind as a candidate for differential diagnosis of C3-positive cases.
Some authors stated that MCE deficiency could be included in neonatal screening programs due to its consistent biochemical profile and potential for dietary and medical treatment. Nevertheless, our opinion is that a decision on this point must be suspended for the moment. In fact, efficacy of medical and dietary treatment in this disorder is unclear.
Round 2
Reviewer 2 Report
Comments and Suggestions for Authors
Before acceptance, add the reason why MUT gene was not analyzed.
Author Response
Reviewer 2#
Before acceptance, add the reason why MUT gene was not analyzed.
Thank you for your comment. We added a statement on this point at page 3.
Answer: We started with the molecular analysis of PCCA, PCCB and then of MCE genes because of the biochemical findings at NBS and at the diagnostic analyses performed at recall.
MMA was not increased at 2 days of life.
At the recall, MMA was normal on DBS and only slightly altered in urine.
Moreover, the increased levels of 3-OH-isovaleric acid prompted us to analyzed PCCA, PCCB and MCE genes before MUT gene.
In fact, while methylcitrate and 3-hydroxypropionic acid are present in both disorders (PA and MUT), 3-hydroxy-isovaleric acid, is not detectable in MUT deficiency.